# Tradeoffs in demographic mechanisms underlie differences in species abundance and stability

Lauren M. Hallett[1], Emily C. Farrer[1,2], Katharine N. Suding[3], Harold A. Mooney[4] & Richard J. Hobbs[5]

Understanding why some species are common and others are rare is a central question in ecology, and is critical for developing conservation strategies under global change. Rare species are typically considered to be more prone to extinction—but the fact they are rare can impede a general understanding of rarity vs. abundance. Here we develop and empirically test a framework to predict species abundances and stability using mechanisms governing population dynamics. Our results demonstrate that coexisting species with similar abundances can be shaped by different mechanisms (specifically, higher growth rates when rare vs. weaker negative density-dependence). Further, these dynamics influence population stability: species with higher intrinsic growth rates but stronger negative density-dependence were more stable and less sensitive to climate variability, regardless of abundance. This suggests that underlying mechanisms governing population dynamics, in addition to population size, may be critical indicators of population stability in an increasingly variable world.

[1] Environmental Studies Program and Department of Biology, University of Oregon, Eugene, OR 97403, USA. [2] Department of Ecology and Evolutionary Biology, Tulane University, New Orleans, LA 70118, USA. [3] Institute of Arctic and Alpine Research, University of Colorado Boulder, Boulder, CO 80309, USA. [4] Department of Biological Sciences, Stanford University, Palo Alto, CA 94305, USA. [5] School of Biological Sciences, University of Western Australia, Crawley, WA 6009, Australia. Correspondence and requests for materials should be addressed to L.M.H. (email: hallett@uoregon.edu)

Understanding differences in species abundances—why some species are common while others are rare—is a foundational goal of ecology[1–3], as enunciated in Preston's 1948 paper "The commonness, and rarity, of species", and is at the heart of most conservation management efforts. Species vary both in their long-term abundances—whether they are, on average, rare or common—and in the degree to which their abundances fluctuate over time. A common assumption for species conservation is that these dynamics are linked, such that large populations are more stable and small populations more variable due to stochasticity in demographic events[4,5]. Some abundant species exhibit strong fluctuations over time, however, while some rare species are strikingly stable[6]. Here we revisit Preston (1948), hypothesizing that variation in the mechanisms that determine average abundance may influence when population size is linked to stability.

Species abundances can be shaped in two main ways. Firstly, individuals often exhibit lower realized growth and survival rates as a species becomes more abundant[7]. This phenomenon, called negative intraspecific density-dependence (NDD), is thought to occur because natural enemies or competitive interactions limit conspecifics more than heterospecifics[8]. NDD can constrain population size such that less self-limited species often have higher average abundances (Fig. 1a)[9–12]. Secondly, when a species is rare, interactions with other species predominate, and individuals often exhibit maximum growth and survival rates[13]. This maximum rate of population increase is set by intrinsic reproductive and death rates, and a high growth rate when rare (GRWR) should contribute to a species' average abundance (Fig. 1b). Theory predicts that NDD and GRWR should be joint determinants of average abundance[9] (Fig. 1c), and integrating these mechanisms is important in understanding species coexistence[14]. However, previous empirical studies of species abundance have focused only on either NDD[10,11,15] or intrinsic growth[16,17].

An integrated understanding of the processes that determine the average abundances of species within a community may explain differences in their population stability (here defined as temporal mean divided by temporal standard deviation)[12]. A common assumption is that large populations are more stable[4,5]. However, high GRWR should promote stability regardless of population size by allowing populations to rapidly recover[12,18] and is mathematically linked to stability[19]. In addition, the degree to which species' population sizes vary over time due to stochastic or environmentally driven variation in growth rate should be dampened by strong NDD[6,20]. For two species with similar average abundances, these effects should combine such that the species characterized by a higher GRWR but stronger NDD is more stable, regardless of its abundance.

We develop and test this framework to understand abundance and stability in relation to population dynamics. We utilize a 32-year demographic record of annual plants with ample data points for species with a range of abundances. The system is suited for our analyses because it is characterized by high rainfall variability and dominated by small statured, highly dynamic annual species with minimal multi-year seed bank carryover[21,22], allowing us to reasonably describe their populations with simple models. First, we test whether either mechanism (GRWR or NDD) alone is sufficient for predicting abundances or whether both are necessary. We find that coexisting species can exhibit similar average abundances through different mechanisms (e.g., weaker NDD vs. higher GRWR). Second, we examine relationships between population stability, GRWR, and NDD. We find that population stability is more strongly governed by the mechanisms underlying a species' average abundance—particularly a higher GRWR that may enhance population recovery—than the resultant population size.

## Results and Discussion

Niche overlap was minimal among all species, and intraspecific NDD appeared to be a stronger determinant of species' growth rates than total density dependence (Supplementary Figs. 3 and 4). This aligns with previous research in our system[13] and others[23] that suggests strong niche differentiation (i.e., stronger intra- than interspecific competition), and supports a focus on GRWR and NDD as determinants of abundance.

Consistent with our expectations, species varied in the demographic parameters that governed their average abundances (Fig. 2a). Neither GRWR ($R^2 = 0.18$, $P = 0.56$) nor NDD ($R^2 = -0.29$, $P = 0.33$) predicted abundance in isolation, but mean abundance was significantly related to the combined effect of both mechanisms (regression analysis; Fig. 2b; Supplementary

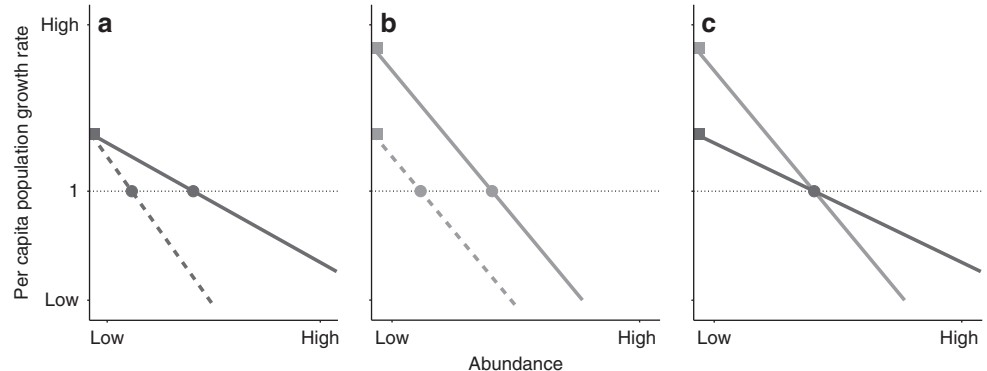

**Fig. 1** A framework to compare mechanisms that shape species abundance. Lines represent species; we assume here that all species have per capita growth rates (calculated as the population size in time $t + 1$ divided by the population size in time $t$) that decrease with abundance. A species' growth rate when rare (GRWR) is indicated by a species' y-intercept (squares). Intraspecific negative density-dependence (NDD) is indicated by the slope of a species' line. We expect dynamics to shift a species abundance toward the case where per capita population growth rate is one (the circles, which we term predicted average abundance). **a** Predicted average abundance decreases with NDD; the solid species is more abundant than the dashed species due to weaker NDD for the solid species. **b** Predicted average abundance increases with GRWR; the solid species is more abundant than the dashed species due to a higher GRWR of the solid species. **c** Two species may be equally abundant but for different reasons; the dark species has weaker NDD whereas the light species has a higher GRWR. The light species can recovery more quickly when rare and should therefore be more stable over time. Note that relationships are not necessarily linear, but are depicted as such for visual simplicity

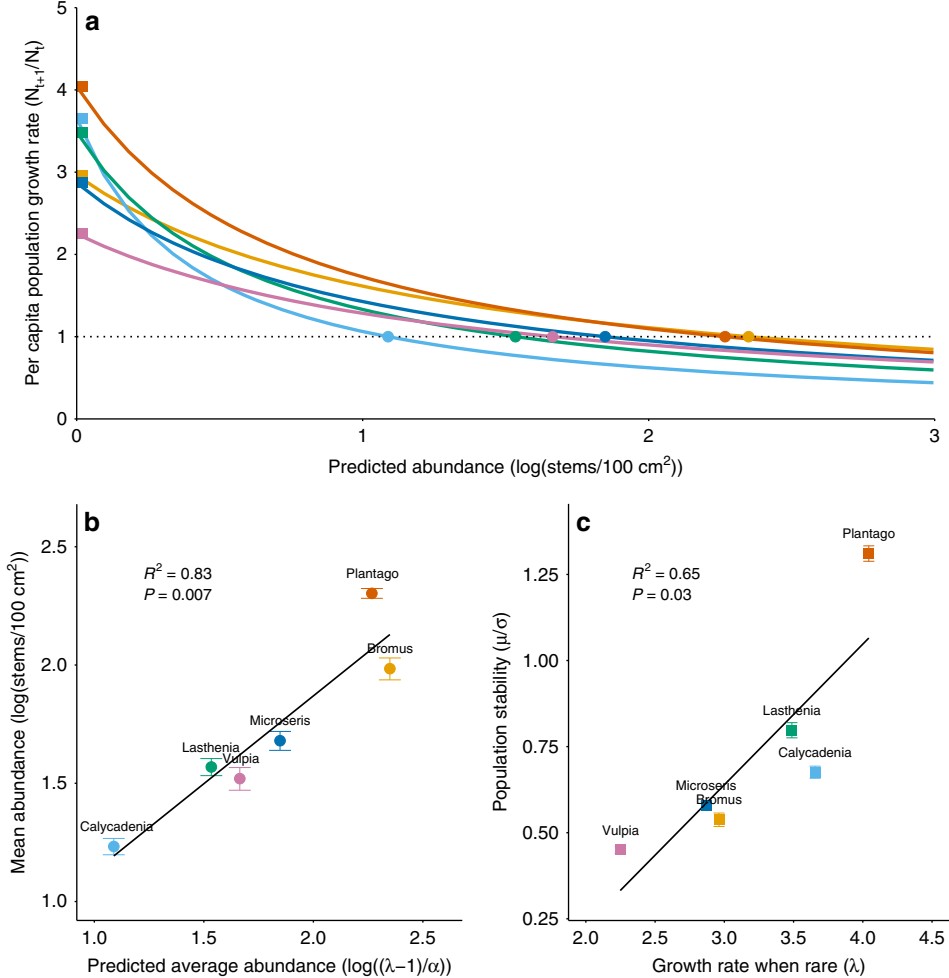

**Fig. 2** Growth rate when rare and negative density dependence shape population size and stability. **a** Relationship between per capita population growth rate and abundance for six serpentine grassland species. Differences in species' average growth rates when rare (GRWR; mathematically indicated by $\lambda$ and visually by squares on the $y$-intercept) and intraspecific negative density-dependence (NDD; mathematically indicated by $\alpha$ and visually by the slope of the line) result in different predicted average abundances (indicated by circles on the dotted line). **b** Observed mean abundance over time (±s.e.m., $n = 150$) in relation to predicted average abundance (circles from panel **a**). **c** Observed species stability over time (±s.e.m., $n = 150$) in relation to GRWR (squares from panel **a**)

Fig. 5a). Pairwise species comparisons highlighted that species governed by different mechanisms can be equally abundant. For example, both *Plantago* and *Bromus* were highly abundant, but for different reasons: *Plantago* was abundant due to its high GRWR and despite moderate NDD, whereas *Bromus* was abundant due to weak NDD and despite a much lower GRWR. Variation among species resulted in a spread in both abundances and associated demographic mechanisms (Fig. 2a; Supplementary Fig. 5a). For example, *Plantago* and *Calycadenia* exhibited the highest GRWR, but differences in their degree of NDD led to very different abundance patterns: *Plantago* was among the most abundant, whereas *Calycadenia*, which exhibited very strong NDD, was the least. These dynamics help to explain why we did not observe individual effects of GRWR and NDD on abundance; however, other studies suggest they would be discernable with a larger sample size[6,10].

While species with small populations are commonly assumed to be less stable and therefore at greater risk of stochastic extinction[4], we found that population stability (i.e., temporal mean divided by standard deviation, $\mu/\sigma$) was not significantly related to either predicted ($R^2 = 0$, $P = 0.57$) or observed ($R^2 = 0.24$, $P = 0.19$) population size via regression analysis. In contrast,

we observed that species that were rare due to strong NDD but that had high GRWR were very stable (Fig. 2c)[12], likely because local populations were able to recover from stochastic population loss. In our dataset, this phenomenon was exemplified by *Calycadenia*, which maintained a small but stable population due to a relatively high GRWR and NDD. Counter-intuitively, *Calycadenia* may therefore be at lower risk of stochastic extinction than species like *Bromus* and *Microseris*, which are more abundant on average but less able to recover when rare. This aligns with recent findings from Yenni et al. (2017)[6], which demonstrate that persistent, rare species are shaped by strong intraspecific NDD that both causes them to be rare but also enables them to rapidly return to equilibrium. Analytically, we found that GRWR was a stronger determinant of stability ($\mu/\sigma$) than NDD. A high GRWR is associated with both a larger population size and enhanced recovery when rare, which should both increase stability, whereas a high NDD is associated with reduced population variability but also reduced population size (Supplementary Fig. 5b).

Environmental variability can drive fluctuations in abundance over time if species' vital rates differ with environmental conditions, a relationship that may be amplified by climate change. Our observed stability values were not well correlated with predicted

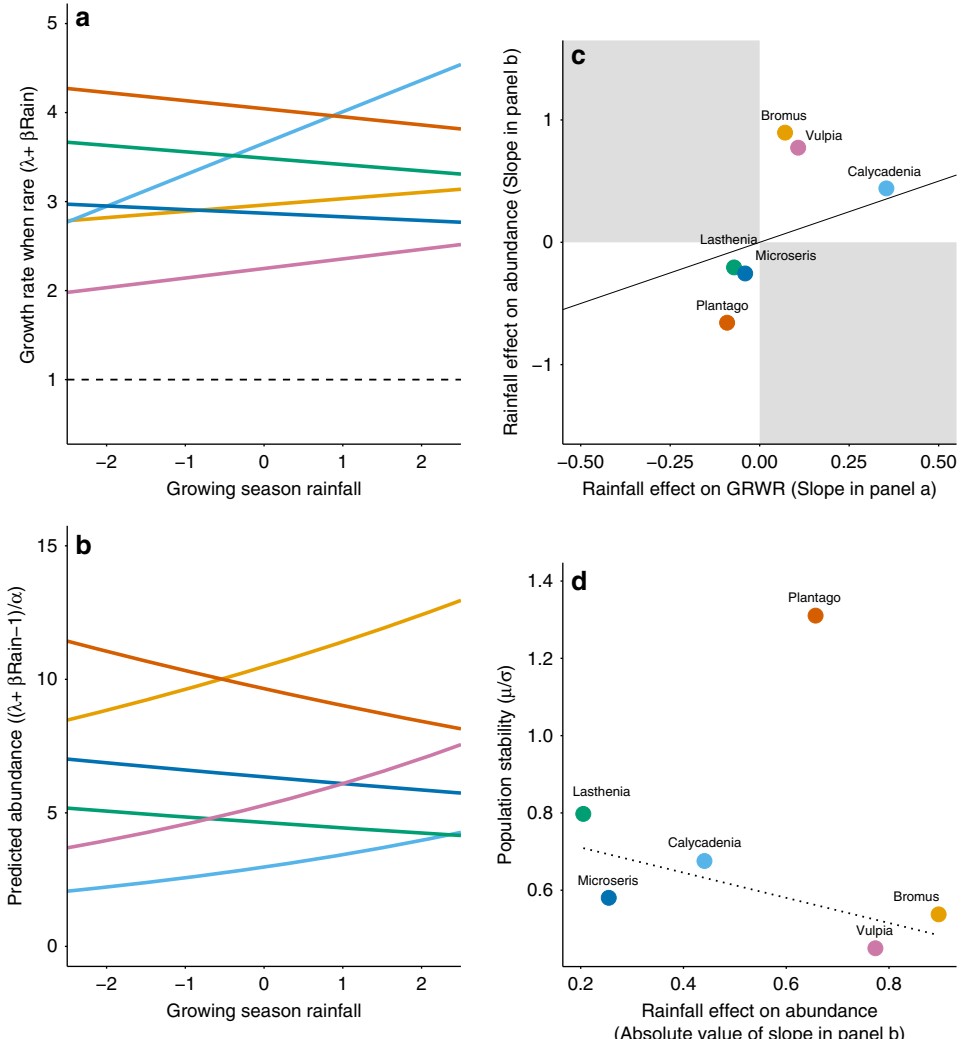

**Fig. 3** Population sensitivity to climate variability depends on abundance mechanisms. **a** Effect of rainfall on the growth rate when rare (GRWR) of six serpentine grassland species. Rainfall is standardized such that 0 represents average conditions. **b** Effect of standardized rainfall on the predicted abundance of each species. **c** Relationship between the GRWR sensitivity to rainfall (slope of panel **a**) and the responsiveness of the population to rainfall (slope of panel **b**); greater NDD dampens population response to rainfall. Line indicates a one-to-one relationship; not-possible areas of the state-space are shaded. **d** Relationship between observed population stability and the responsiveness of the population to rainfall (absolute value of the slope of panel **b**). Non-significant regression lines omits the dominant species *Plantago* ($R^2 = 0.42$, $P = 0.14$)

stability. Species' sensitivities to rainfall variability likely contributed to this disconnect. In our study, the direct effect of rainfall on GRWR varied considerably, with some species, such as *Microseris*, showing minimal sensitivity, and others, such as *Calycadenia*, showing large shifts in annual GRWR with rainfall (Fig. 3a). However, the magnitude of species' predicted population responses (Fig. 3b) was jointly affected by a rainfall effect on GRWR and by a species' degree of NDD (Fig. 3c). Consequently, *Calycadenia* exhibited minimal response to rainfall at the population level, whereas the two grasses, *Bromus* and *Vulpia*, exhibited rainfall-driven shifts in predicted population size despite moderate sensitivity in GRWR (Fig. 3c). In general, large predicted shifts in population size were negatively associated with observed stability (the exception being the dominant, very stable *Plantago*; Fig. 3d). This suggests that species like *Calycadenia* may be stabilized in multiple ways: their populations can recover quickly due to high GRWR, and their population response to environmental variability is minimized by strong NDD, an effect not captured by our analytical predictions that assume average rainfall. Moreover, while we focused exclusively on the direct

effects of rainfall, Chu et al. (2016)[23] found that, across species with a range of abundances and demographic parameters, indirect effects of climate variability were also minimized for species with strong intraspecific NDD.

Our study demonstrates that population stability is more closely related to the mechanisms governing population dynamics than to the resultant population size. Integrating GRWR and NDD is central to understanding species coexistence[7,13,14], and our results indicate this integration is also essential to predicting both population size and stability. To our knowledge it is the first empirical test of these relationships, in part due to the intensive data collection required to parameterize models for uncommon species[10]. While our model reflects some realism in the factors governing population dynamics —non-linear relationships between growth rate and abundance, precipitation effects on GRWR—it is unquestionably a simple model. Modifications would be necessary to explore more complex relationships with processes that influence dynamics over broader spatial and temporal scales, notably, inclusion of Allee effects and the effect of disturbance.

Our approach provides insights for conservation and management decisions under global change. Conservation efforts frequently use population size as a benchmark for assessing species' vulnerability to extinction[24]. In terms of rare species conservation, our results indicate that greater intervention may be necessary to maintain species that are less abundant due to low GRWR as opposed to strong NDD. Similarly, our findings indicate that opportunities for invasive species management may vary with time. Counter to the common perception that abundant invasive species should be prioritized, our study suggests that interventions may be most efficacious when both the abundance and GRWR of the non-native species are low (such as at the end of several dry years for *Bromus*). Experiments that parameterize GRWR and NDD are necessary to test the generality of our results to other systems and to explore general patterns between demographic rates, abundances, and stability.

## Methods

**Data collection**. Our study is based in a serpentine grassland at the Jasper Ridge Biological Preserve in San Mateo County, California, USA (122º12′ W, 36º25′N). Soils at the site are characteristically shallow (<40 cm deep), with low nutrient concentrations, high Ni and Mn concentrations, and a low Ca:Mg ratio. The site experiences a Mediterranean climate with cool, wet winters and hot, dry summers with five-fold variation in annual rainfall[25]. Mean growing season rainfall (September–April) over the study period was 604 mm but varied greatly across that time period, from 228 to 1155 mm. The site is dominated by annual plants (primarily annual forbs and a few annual grasses) that germinate in autumn and set seed in spring and summer (Supplementary Table 1; Supplementary Fig. 1a). Previous research indicates that species at the site have minimal multi-year seed bank carryover (Supplementary Methods)[21].

Each April from 1984 to 2015 we censused stem counts by species in permanently marked 10 × 10 cm plots. This plot size was selected on the basis of the small stature of the annual plants and the high plant densities common at the site (several thousand plants per m²)[21]. An initial set of 30 plots was set up in 1983 (ref. [21]). Periodic gopher disturbance is common at the site. To ensure that post-gopher successional trajectories were stratified across years we added at least 10 additional plots on fresh gopher mounds every year between 1987 and 1996. We also recorded any fresh gopher disturbance in the plots. This resulted in a total of 150 replicate plots whose time series ranged from 20 to 32 years.

We utilized daily precipitation records from Jasper Ridge to characterize rainfall during the study period. Particularly in the early years of the study there was some missing data in the Jasper Ridge record. When missing values occurred we substituted precipitation data from the Woodside Fire Station (National Center for Environmental Information, ID GHCND:USC00049792), located 3 km to the northwest at 116 m elevation, standardized against existing Jasper Ridge records. Rainfall data are presented as growing season rainfall (i.e., from September to April) (Supplementary Fig. 1b).

**Data selection**. We restricted analyses to species with a mean population density of ≥1 individual/100 cm² over the entire site and sampling period. This resulted in a set of six focal species: four native annual forbs (*Calycadenia multiglandulosa, Lasthenia californica, Microseris douglasii, Plantago erecta*), a native annual grass (*Vulpia microstachys*), and a non-native annual grass (*Bromus hordeaceus*) (Supplementary Table 1 and Supplementary Fig. 1a). These six species exhibited a range in average abundance and population stability[26] (Supplementary Table 1 and Supplementary Fig. 1). We removed data points that experienced gopher disturbance in either the current or previous time step; total stem density was fairly constant after these points were removed (Supplementary Fig. 2).

We described population size for each species as the temporal mean number of individuals when the species was present, calculated within a plot and then averaged across plots. Second, we described population stability for each species as the temporal mean divided by standard deviation ($\mu/\sigma$)[26], calculated within a plot and then averaged across plots, excluding plots in which the focal species never occurred. All analyses throughout were conducted in R version 3.4.1.

**Model selection**. We considered five candidate models that parameterize population $N$ of species $i$ at time $t + 1$ as a function of the population in the previous time step, $N_{i,t}$, GRWR ($\lambda_i$), intraspecific competition (i.e., NDD $\alpha_{ii}$), and interspecific competition ($\alpha_{ij}$) (models adapted from Law and Watkinson's analysis of annual plant competition;[27] Supplementary Table 2). Candidate models reflected scenarios in which NDD was both linear and non-linear (i.e., in which growth rate linearly and non-linearly declined with abundance). Because rainfall is highly variable at the site, we included an additional term ($\beta_i$) to quantify the degree to which growing season rainfall modifies a species' annual GRWR. We standardized growing season rainfall around 0, such that 0 represented an average rainfall year. We logged species counts and used maximum likelihood to fit each model for each

of the six species and compared the model fits using AICs summed across all six species models. The best-fit model is commonly used for quantifying plant population dynamics and species coexistence (Supplementary Tables 2 and 3)[28]

$$N_{i,t+1} = \frac{(\lambda_i + \beta_i \text{Rain}_{t+1})N_{i,t}}{1 + \alpha_{ii}N_{i,t} + \sum_1^{j \neq i} \alpha_{ij}N_{j,t}}. \tag{1}$$

Previous research in our system suggests it is shaped by strong niche differentiation[13]. To examine this assumption, we first visualized per capita population growth rate (calculated as the change in stem count between time $t$ and time $t + 1$) in relation conspecific and summed heterospecific abundances at time $t$. This visualization highlights that growth rates consistently decrease with conspecific abundance but not summed heterospecific abundance for all species in our analyses (Supplementary Figure 3). In other words, intraspecific NDD appears to be a stronger determinant of species' growth rates than total density dependence.

Second, we used our population models to parameterize the degree of niche overlap between species. We calculated niche overlap as

$$\rho = \sqrt{\frac{\alpha_{ij}}{\alpha_{jj}} \frac{\alpha_{ji}}{\alpha_{ii}}}, \tag{2}$$

where complete niche differentiation is indicated by $\rho = 0$, and complete niche overlap is indicated by $\rho = 1$ (refs [29,30]).

We calculated $\rho$ for every pairwise species combination from the best-fit model; in all instances niche overlap was minimal ($\rho \leq 0.25$ for every pairwise species combination; Supplementary Figs 3 and 4). As a result, we simplified the models by removing interspecific competition terms, such that a species' population $N$ at time $t + 1$ is solely a function of the population in the previous time step, $N_t$, modified by GRWR $\lambda$, rainfall response $\beta$, and intraspecific density dependence $\alpha$. The same model structure again had the lowest AIC (Supplementary Table 4). Throughout we use parameter estimates from this best-fit model (Supplementary Table 5), which is simplified by the removal of interspecific competition terms to

$$N_{t+1} = \frac{(\lambda + \beta \text{Rain}_{t+1})N_t}{1 + (\alpha N_t)}, \tag{3}$$

where a species' population $N$ at time $t + 1$ is a function of the population in the previous time step, $N_t$, modified by GRWR $\lambda$, rainfall response $\beta$, and self-limitation $\alpha$. Thus, a species equilibrium abundance in any given year is calculated as $(\lambda + \beta \text{Rain} - 1)/\alpha$, and its average abundance over time predicted by $(\lambda - 1)/\alpha$[14].

**Analyses**. We used the fitted parameters to test our prediction that species governed by different processes can be equally abundant. To test whether GRWR or NDD alone could predict average abundance, we regressed observed mean abundance against $\lambda$ alone and $\alpha$ alone. To test whether the combination of mechanisms explained abundance we regressed mean abundance by predicted average abundance, calculated as $(\lambda - 1)/\alpha$. To an extent this amounts to an additional test of model fit, but it provides a comparison point for each mechanism in isolation and is essential to confirm prior to interpreting parameter estimates. For this analysis we removed the effect of environmental variability by setting the standardized rainfall input $\text{Rain}_{t+1} = 0$, reflecting an 'average' rainfall year. To generalize our findings, we calculated predicted equilibrium abundance across the range of GRWR ($\lambda$) and NDD ($\alpha$) parameter estimates from our population models.

We next tested the hypothesis that population stability is determined by population dynamics mechanisms (rather than abundance per se). First, we regressed population stability against abundance (predicted and observed) and GRWR (again setting $\text{Rain}_{t+1} = 0$). Second, we solved for stability, assuming a Poisson distribution and average rainfall conditions, as

$$\frac{\mu}{\sigma} = \frac{(\lambda - 1)/\alpha}{\sqrt{(\lambda - 1)/\alpha}}. \tag{4}$$

We used this equation to calculate predicted stability across the range of GRWR ($\lambda$) and NDD ($\alpha$) parameter estimates from our population models.

We assessed the sensitivity of species' abundances to rainfall by calculating the predicted equilibrium abundance $(\lambda + \beta \text{Rain}_{t+1} - 1)/\alpha$ across the range of observed rainfall values. We qualitatively compared the slope of this relationship (i.e., the change in predicted equilibrium abundance by rainfall) with the slope of the relationship between annual GRWR and rainfall (i.e., $\beta$) to describe the joint effect of rainfall response and NDD on variation in predicted equilibrium abundance. Finally, we regressed observed population stability against the absolute value of a predicted effect of rainfall on abundance.

## Data availability

The data underlying this study are available from the corresponding author upon reasonable request.

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

## Acknowledgements

We thank David Ackerly, Nick Kortessis, Ryan Langendorf, Max Li, Peter Ralph, Lauren Shoemaker, Marko Spasojevic, and the Suding Lab for their comments. We thank the Jasper Ridge Biological Preserve for permission to continue work there. Initial funding support came from a NATO postdoctoral fellowship to R.J.H., subsequent funding has included support from the National Science Foundation, Mellon Foundation, CSIRO, Murdoch University, and the ARC Centre of Excellence for Environmental Decisions.

## Author contributions

R.J.H., H.A.M., and L.M.H. collected the data; L.M.H. and K.N.S. developed the idea; L.M.H. and E.C.F. conducted the analyses; L.M.H. wrote the manuscript; R.J.H., H.A.M., E.C.F., and K.N.S. edited it.

## Additional information

**Competing interests:** The authors declare no competing interests.

