## [Peer Review File · Nature Communications]

Reviewers' comments:

Reviewer #1 (Remarks to the Author):

The work examines the relative abundances of plant species in a long-term study in California focusing on the potentially dual roles of self regulation and population growth response when rare. The work builds off of theoretical work (e.g. Chisholm) and speaks to some recent findings of great interest in plant ecology concerning the strength of intra-specific negative density dependence as it relates to abundance (e.g. Comita et al.). Thus, the work is timely, concerns a topic of general interest and nicely blends theory with detailed long-term population data that is required for a convincing study.

The authors find that neither self limitation or growth when rare alone explain expected population sizes, but their combined effects can predict population sizes. Furthermore, the authors show species with similar population sizes can indeed arrive at that size via different combinations of the 2 mechanisms. These are exciting and important results. I think the work will influence the thinking of those focusing primarily on density dependence and co-existence.

The analyses and data are appropriate for the study and the inferences made a largely appropriate. I have, really, only two minor comments that I think are worth considering.

First, the model states that self-limitation (α) is a competition parameter. I think this is unnecessarily narrow. Other factors such as shared enemies could possibly cause this limitation. The authors may have a good reason to believe this is not the case in their system, but I think generalizing that parameter to being open to other possibilities would be useful in the literature. I'm only suggesting a minor change on line 89 to deal with this issue.

Second, while they are not 'significant' the R^2 values on a few of those analyses are fairly high (lines 97-98). The lack of significance underlies one of the main interesting punchlines here - that only the combination of forces dictates abundance. My question is whether the authors think the lack of significance is primarily due to only having a few species in the study. That is, would a study with many more species likely find similar variance explained and a "significant" result for single mechanism analyses? I suspect it might. That would not undercut this paper in my mind as in this imaginary second study the combined effect presumably would still be more informative. I'm wondering if the authors can speak to this in a response and perhaps provide a sentence in the MS as well on this issue.

Otherwise, I look forward to seeing this work in print and I imagine it will nicely influence our field.

Signed,
Nate Swenson

Reviewer #2 (Remarks to the Author):

Hallett et al. present a study of population dynamics for 6 species of annual plants for over 30 years to test mechanisms of species rarity and stability. The study emphasizes the combined role of stabilizing and equalizing processes (called "self-limitation" and "growth rate when rare" by the authors respectively) to determining species rarity and stability. They also find a role of climate variability in determining population dynamics over this long time-frame. I found this study appropriate, novel and timely – it should be of broad interest to the readers of Nature Communications. I had only a few minor suggestions that I hope will improve the manuscript and broaden its appeal by clarifying a few key results/issues.

Lines 44-55: This is all accurate, but my concern is that your terminology here might confuse some readers. What you refer to as “self-limitation” (what Chesson referred to as “stabilizing processes”) is indeed lower growth rates at higher abundances, but the inverse of self-limitation, i.e. higher growth rates at lower abundances, is very similar to your terminology for what Chesson referred to as equaling mechanisms (what you call “high growth rate when rare” or “GRWR”). This is all clear when one reads the Introduction thoroughly, but my fear is that many readers may not be that thorough, and the distinction here is not entirely clear from the abstract alone. I'm not suggesting using Chesson's terminology (which, in my opinion, is equally prone to confusion), but rather considering alternative terminology that might make the distinction here clearer. Just to clarify, on a plot with per-capita growth rate on the y-axis, and species abundance (or frequency) on the x-axis, what you call “self-limitation” would be the slope of the line and what you call “GRWR” would be the y-intercept (This is what you state in Fig. 1). If that is the case, “GRWR” more accurately reflects not just the population growth rate when rare, but also the average population growth rate across all abundances (i.e. if the “y-intercept” were not at frequency = 0.0, but instead at frequency = 0.5). In this sense, you might consider changing “high growth rate when rare” to “high average growth rate”, and then you might consider changing “self-limitation” to “frequency-dependence” or “density-dependence” (as it is referred to by Vellend [Princeton Monographs, 2016] and others).

Lines 47-48: Again, one could think of this as self-limitation, which reflects the decline in population growth rates at high abundances. But frequency- or density-dependence (i.e. self-limitation) also reflects the degree to which population growth rates increase as a species becomes more rare. In my opinion, this density- or frequency-dependence in population growth rates is lost by referring to it as “self-limitation.”

Lines 56-64: The statements in this paragraph partially depend on how one defines stability. Stability defined as absolute variability in population abundance over time (i.e. SD of population abundance over time) should have more to do with what you refer to as “self-limitation” (i.e. the slopes of the lines in Fig. 1) than “GRWR” (i.e. the y-intercepts of the lines in Fig. 1). In other words, while both the gray and black species in Fig. 1c should have the same equilibrium abundances, all else being equal, the gray species (with stronger self-limitation) would likely have greater stability given that they would have greater increases/decreases in their population growth rates as their abundance decreases/increases. However, I understand that you may not be defining stability in this way, and instead defining it as the ability to stay common or rare in a more categorical fashion. Below, you refer to stability as the CV of population abundance through time. In that sense, you may be correct here, but it is unclear to me. I would recommend at least explicitly defining what you mean by stability here. Regardless of which influences stability, I do completely agree that self-limitation and GRWR combine to determine a species average abundance (which seems to be your broader point here).

Lines 112-113: It is not that clear to me from the results presented here or in Fig. 2c that species with stronger self-limitation were more stable (after accounting for GRWR). I can see from looking at the Extended data (Table 5) that *Vulpia*, *Bromis*, and *Microseris* had the lowest self-stabilization and were relatively unstable despite high mean abundance for *Bromus* and moderate abundance for *Vulpia* and *Microseris*. Could you include an additional analysis to support this sentence, which tests for an influence of self-limitation on stability after controlling for GRWR (e.g. partial regression analysis)?

Fig. 2: I would recommend making it more explicit in the figure caption that λ and α relate to GRWR and self-limitation (respectively). This is clear in the text, but not explicit in the figure or caption. Otherwise, it's not clear from this figure or caption alone that both processes combine to influence mean abundance.

Reviewer #3 (Remarks to the Author):

The authors fit population dynamical models to long-term time series data for 6 annual plant species growing in a serpentine grassland. They ask why the rare species are rare: because they have lower maximum per-capita growth rates than common species, experience stronger intraspecific competition, or some combination? They also ask if stronger intraspecific competition stabilizes species' dynamics in the face of year-to-year variation in rainfall that alters species' maximum per-capita growth rates. They find that maximum per-capita growth rate and strength of intraspecific competition vary independently of one another, so that different species are rare (or common) for different reasons. Species also varied in their stability, which was not correlated with their mean abundances and not determined solely by the sensitivity of their maximum per-capita growth rates to rainfall.

I enjoyed the ms. It uses the right tools for the job--parameterizing a population dynamic model and then using the parameter estimates to answer the ecological questions of interest. The analyses are straightforward and easy to understand (though I do have some suggestions for analytical improvement; I think the authors could take even fuller advantage of having a parameterized population dynamic model). My main concern is that it's an analysis of one community and I'm not sure how much it adds to recent similar analyses of many communities (Yenni et al. 2017 *Global Ecol Biogeog*, Chu et al 2016 *Nature Commun*).

Detailed comments:

1. lines 38-39: I question whether it's a "common assumption" that abundant species have more stable population dynamics than rare ones. Depends on the measure of stability, but the authors freely switch back and forth between using "stability" to refer to different things that are expected to behave in different ways. It's well-established that the temporal variance of population size scales with mean population size as a power law with an exponent between 1 and 2, a phenomenon known as Taylor's law (Taylor 1961; reviewed in Reumann et al. 2017 *PNAS*; see also various recent papers by Joel Cohen). So if your measure of stability is the temporal variance or standard deviation of abundance, the "common assumption" is that abundant species will vary more than rare ones and thus be less stable. But if your measure of stability is the coefficient of variation (standard deviation divided by the mean), larger populations will be more stable. Alternatively, your measure of stability might be the rate of return to equilibrium, or the rate of return to the stationary distribution in the case of a stochastic model; the latter has a complicated relationship to the temporal variance in some cases (Ziebarth et al. 2010 *EcoLetts*). Finally, if your measure of stability is extinction risk (risk of dropping to zero abundance within some specified time frame), then typically rare species will be at greatest risk even if they're more stable than common ones according to other measures of stability such as temporal variance, temporal standard deviation, or cv. Indeed, the fact that rare species are at greater risk of extinction than common ones (and so "less stable" by that measure) is one reason why we might expect the rare species we observe to exhibit *more* stable dynamics than common species, in the sense of bouncing back to equilibrium faster than common species (Yenni et al. 2012 *Ecology*, 2017 *Global Ecol Biogeog*). Any rare species that don't bounce back quickly following perturbation are likely to go extinct and so not last long enough to be observed. I suggest that the authors make clearer throughout the ms what measure of stability is of interest, and why.

2. A suggestion/question: why use multiple regression to ask whether variation in maximum per-capita growth rate, strength of density-dependence, or both predict mean abundances and population cv? You have a fitted population dynamic model that seems to fit well, at least as far as can be judged by the information provided. Can't you just do a Taylor series expansion around equilibrium population size for each species in an average rainfall year and figure out how the equilibrium population size and the variance around the equilibrium depend on the model parameters alpha, lambda, and beta? That gives you a lot more information than the linear regressions, doesn't it? For instance, it tells about "interaction terms"--how the effects of one model parameter depend on the values of the other parameters. It lets you generalize your

conclusions beyond these 6 species to any species to which this model applies. You could solve for rate of return to equilibrium, and prove that it depends on both maximum per-capita growth rate and strength of intraspecific competition. Etc.

3. Lines 135-137: I was surprised not to see even brief discussion of the results of Yenni et al. (2017 *Global Ecol Biogeog*). They use methods similar to those used in this ms to show for 90 communities of various sorts of species (rather than just one plant community) that rare species almost invariably experience stronger negative frequency dependence than common ones. This implies that persistent rare species are both rare and persistent because they experience much stronger intra- than interspecific competition than common species. That strong intraspecific competition (relative to interspecific competition) both makes rare species rare and helps them bounce back quickly to equilibrium.

4. I suggest adding a discussion relating the results to those of Peter Adler's group. They find that in several US perennial plant communities, intraspecific competition is strong relative to interspecific competition. As a result, the responses of species' abundances to abiotic environmental variation don't depend much on interspecific interactions, except when the species within a community differ greatly in their responses to the abiotic environment. See, e.g., Chu and Adler 2015 *Ecology* and Chu et al. 2016 *Nature Comm*. And although they don't mention it, it's clear from their data (e.g., Fig. 3 in Chu et al. 2016) that species' abundances depend on both their maximum per-capita growth rates and on the strength of intraspecific competition they experience (relative to the strength of interspecific competition).

5. Just a suggestion: instead of Fig. 3, I'd rather see a plot of the temporal variance of each species' abundance vs. the temporal variance of its maximum per-capita growth rate. That seems like a more easily-interpretable way to visualize which species are stable because they're insensitive to rainfall variation, and which species are stable because they are sensitive to rainfall variation but damp out its effects with strong density dependence. Ziebarth et al. (2010 *EcoLetts*) emphasize the value of this sort of analysis and show that many species seem to have only a weak tendency to bounce back from perturbations.

Reviewer #1 (Remarks to the Author):

The work examines the relative abundances of plant species in a long-term study in California focusing on the potentially dual roles of self regulation and population growth response when rare. The work builds off of theoretical work (e.g. Chisholm) and speaks to some recent findings of great interest in plant ecology concerning the strength of intra-specific negative density dependence as it relates to abundance (e.g. Comita et al.). Thus, the work is timely, concerns a topic of general interest and nicely blends theory with detailed long-term population data that is required for a convincing study.

The authors find that neither self limitation or growth when rare alone explain expected population sizes, but their combined effects can predict population sizes. Furthermore, the authors show species with similar population sizes can indeed arrive at that size via different combinations of the 2 mechanisms. These are exciting and important results. I think the work will influence the thinking of those focusing primarily on density dependence and co-existence.

The analyses and data are appropriate for the study and the inferences made a largely appropriate. I have, really, only two minor comments that I think are worth considering.

Thank you for your positive feedback! We are glad that you also find the study exciting.

First, the model states that self-limitation (α) is a competition parameter. I think this is unnecessarily narrow. Other factors such as shared enemies could possibly cause this limitation. The authors may have a good reason to believe this is not the case in their system, but I think generalizing that parameter to being open to other possibilities would be useful in the literature. I'm only suggesting a minor change on line 89 to deal with this issue.

Makes sense, we have changed the line to "intraspecific density dependence".

Second, while they are not 'significant' the R2 values on a few of those analyses are fairly high (lines 97-98). The lack of significance underlies one of the main interesting punchlines here - that only the combination of forces dictates abundance. My question is whether the authors think the lack of significance is primarily due to only having a few species in the study. That is, would a study with many more species likely find similar variance explained and a "significant" result for single mechanism analyses? I suspect it might. That would not undercut this paper in my mind as in this imaginary second study the combined effect presumably would still be more informative. I'm wondering if the authors can speak to this in a response and perhaps provide a sentence in the MS as well on this issue.

This is an interesting point. We think it is a combination – because α and λ were not well correlated, species with comparable α s could have different average abundances (as could species with comparable λ s). Because of this, while the general trend was for large λ s and small α s to increase abundance, without accounting for both terms in the regression there is a lot of noise. However – if we had a much larger sample size, it is possible that the effect for each term would be significant independently. We have added the following sentence to the MS on this issue, and include visuals for the reviewer below relating each individual term with species abundance.

L125 “These dynamics help to explain why we did not observe individual effects of GRWR and NDD on abundance, however other studies suggest they would be discernable with a larger sample size^{1,2}.”

Citations: Comita et al 2010, Yenni et al 2017

Otherwise, I look forward to seeing this work in print and I imagine it will nicely influence our field.

Signed,
Nate Swenson

Reviewer #2 (Remarks to the Author):

Hallett et al. present a study of population dynamics for 6 species of annual plants for over 30 years to test mechanisms of species rarity and stability. The study emphasizes the combined role of stabilizing and equalizing processes (called “self-limitation” and “growth rate when rare” by the authors respectively) to determining species rarity and stability. They also find a role of climate variability in determining population dynamics over this long time-frame. I found this study appropriate, novel and timely – it should be of broad interest to the readers of Nature Communications. I had only a few minor suggestions that I hope will improve the manuscript and broaden its appeal by clarifying a few key results/issues.

Thank you for your kind words. We appreciate your interest and suggestions for improvement.

Lines 44-55: This is all accurate, but my concern is that your terminology here might confuse some readers. What you refer to as “self-limitation” (what Chesson referred to as “stabilizing processes”) is indeed lower growth rates at higher abundances, but the inverse of self-limitation, i.e. higher growth rates at lower abundances, is very similar to your terminology for what Chesson referred to as equalizing mechanisms (what you call “high growth rate when rare” or “GRWR”). This is all clear when one reads the Introduction thoroughly, but my fear is that many readers may not be that thorough, and the distinction here is not entirely clear from the abstract alone. I’m not suggesting using Chesson’s terminology (which, in my opinion, is equally prone to confusion), but rather considering alternative terminology that might make the distinction here clearer. Just to clarify, on a plot with per-capita growth rate on the y-axis, and species abundance (or frequency) on the x-axis, what you call “self-limitation” would be the slope of the line and what you call “GRWR” would be the y-intercept (This is what you state in Fig. 1). If that is the case, “GRWR” more accurately reflects not just the population growth rate when rare, but also the average population growth rate across all abundances (i.e. if the “y-intercept” were not at frequency = 0.0, but instead at frequency = 0.5). In this sense, would might consider changing “high growth rate when rare” to “high average growth rate”, and then you might consider changing “self-limitation” to “frequency-dependence” or “density-dependence” (as it is referred to by Vellend [Princeton Monographs, 2016] and others).

Thanks for your thoughts on how to be most clear, we also struggled with terminology. In our initial draft we used negative density-dependence (NDD), and then switched to self-limitation to avoid acronyms, to be more accessible to a wide audience, and to be clear that it is intraspecific limitation. We see the pros and cons of both, but have returned to using NDD based on your thoughts and because it is more commonly used.

If the relationship between abundance and growth rate was linear, then we agree that GRWR and average population growth rates would be effectively the same thing. However, the best-fit model for the data was non-linear, and so average population growth rate and GRWR are not necessarily the same (i.e., for species with equal lambda [GRWR], the species with the smallest alpha [NDD] would have a higher average growth rate). As such, we have retained the GRWR terminology. We have included a line in the figure caption that, although we use a linear model for figure 1 for visual simplicity, the relationship may be non-linear.

Lines 47-48: Again, one could think of this as self-limitation, which reflects the decline in population growth rates at high abundances. But frequency- or density-dependence (i.e. self-limitation) also reflects the degree to which population growth rates increase as a species becomes more rare. In my opinion, this density- or frequency-dependence in population growth rates is lost by referring to it as “self-limitation.”

That makes sense, we have updated to negative density-dependence.

Lines 56-64: The statements in this paragraph partially depend on how one defines stability. Stability defined as absolute variability in population abundance over time (i.e. SD of population abundance over time) should have more to do with what you refer to as “self-limitation” (i.e. the slopes of the lines in Fig. 1) than “GRWR” (i.e. the y-intercepts of the lines in Fig. 1). In other words, while both the gray and black species in Fig. 1c should have the same equilibrium abundances, all else being equal, the gray species (with stronger self-limitation) would likely have greater stability given that they would have greater increases/decreases in their population growth rates as their abundance decreases/increases. However, I understand that you may not be defining stability in this way, and instead defining it as the ability to stay common or rare in a more categorical fashion. Below, you refer to stability as the CV of

population abundance through time. In that sense, you may be correct here, but it is unclear to me. I would recommend at least explicitly defining what you mean by stability here. Regardless of which influences stability, I do completely agree that self-limitation and GRWR combine to determine a species average abundance (which seems to be your broader point here).

Thanks for this observation, which was also pointed out by reviewer 3. We measure stability as the temporal mean/temporal standard deviation, and that definition holds for the points made in this paragraph. We have clarified our definition of stability and further explored the theoretical relationship between GRWR, NDD and stability by solving for predicted stability analytically (see below).

Lines 112-113: It is not that clear to me from the results presented here or in Fig. 2c that species with stronger self-limitation were more stable (after accounting for GRWR). I can see from looking at the Extended data (Table 5) that *Vulpia*, *Bromis*, and *Microseris* had the lowest self-stabilization and were relatively unstable despite high mean abundance for *Bromus* and moderate abundance for *Vulpia* and *Microseris*. Could you include an additional analysis to support this sentence, which tests for an influence of self-limitation on stability after controlling for GRWR (e.g. partial regression analysis)?

This is a good point that was also raised by reviewer 3. Given our relatively small number of species, we were hesitant to include too many parameters in the regression. Reviewer 3 gave an interesting suggestion of using a approximating the relationship between GRWR/ λ , NDD/ α and stability. We have adopted the spirit of this suggestion, analytically solving for predicted stability under average rainfall conditions. We think this helps to understand how stability changes in relation to the two parameters and generalizes beyond our focal species to any general species for which our model applies.

Fig. 2: I would recommend making it more explicit in the figure caption that λ and α relate to GRWR and self-limitation (respectively). This is clear in the text, but not explicit in the figure or caption. Otherwise, it's not clear from this figure or caption alone that both processes combine to influence mean abundance.

Thanks for this suggestion, which we think substantially increases the figure's accessibility. We have revised the caption as follows:
"Differences in species' growth rates when rare (GRWR; mathematically indicated by λ and visually by squares on the y-intercept) and negative density-dependence (NDD; mathematically indicated by α and visually by the slope of the line) result in different predicted average abundances (indicated by circles on the dotted line)."

Reviewer #3 (Remarks to the Author):

The authors fit population dynamical models to long-term time series data for 6 annual plant species growing in a serpentine grassland. They ask why the rare species are rare: because they have lower maximum per-capita growth rates than common species, experience stronger intraspecific competition, or some combination? They also ask if stronger intraspecific competition stabilizes species' dynamics in the face of year-to-year variation in rainfall that alters species' maximum per-capita growth rates. They find that maximum per-capita growth rate and strength of intraspecific competition vary independently of one another, so that different species are rare (or common) for different reasons. Species also varied in their stability, which was not correlated with their mean abundances and not determined solely by the sensitivity of their maximum per-capita growth rates to rainfall.

I enjoyed the ms. It uses the right tools for the job--parameterizing a population dynamic model and then using the parameter estimates to answer the ecological questions of interest. The analyses are straightforward and easy to understand (though I do have some suggestions for analytical improvement; I think the authors could take even fuller advantage of having a parameterized population dynamic model). My main concern is that it's an analysis of one community and I'm not sure how much it adds to recent similar analyses of many communities (Yenni et al. 2017 *Global Ecol Biogeog*, Chu et al 2016 *Nature Commun*).

Thank you for your overall positive assessment and suggestions for improvement. We think the approach and framing are distinct, although complementary, to Yenni et al 2017 and Chu et al 2016. We have better contextualized our study in relation to those papers (see below).

Detailed comments:

1. lines 38-39: I question whether it's a "common assumption" that abundant species have more stable population dynamics than rare ones. Depends on the measure of stability, but the authors freely switch back and forth between using "stability" to refer to different things that are expected to behave in different ways. It's well-established that the temporal variance of population size scales with mean population size as a power law with an exponent between 1 and 2, a phenomenon known as Taylor's law (Taylor 1961; reviewed in Reumann et al. 2017 PNAS; see also various recent papers by Joel Cohen). So if your measure of stability is the temporal variance or standard deviation of abundance, the "common assumption" is that abundant species will vary more than rare ones and thus be less stable. But if your measure of stability is the coefficient of variation (standard deviation divided by the mean), larger populations will be more stable. Alternatively, your measure of stability might be the rate of return to equilibrium, or the rate of return to the stationary distribution in the case of a stochastic model; the latter has a complicated relationship to the temporal variance in some cases (Ziebarth et al. 2010 EcoLetts). Finally, if your measure of stability is extinction risk (risk of dropping to zero abundance within some specified time frame), then typically rare species will be at greatest risk even if they're more stable than common ones according to other measures of stability such as temporal variance, temporal standard deviation, or cv. Indeed, the fact that rare species are at greater risk of extinction than common ones (and so "less stable" by that measure) is one reason why we might expect the rare species we observe to exhibit *more* stable dynamics than common species, in the sense of bouncing back to equilibrium faster than common species (Yenni et al. 2012 Ecology, 2017 Global Ecol Biogeog). Any rare species that don't bounce back quickly following perturbation are likely to go extinct and so not last long enough to be observed. I suggest that the authors make clearer throughout the ms what measure of stability is of interest, and why.

Thank you for this feedback, which was also pointed out by reviewer 2. In the MS we are defining stability as temporal mean/temporal standard deviation, which can be affected by population size, deviation from the mean, and often rate of return. We have clarified this by defining stability earlier in the introduction. In this initial introduction, we have specified that a relationship between population size and stability is a common assumption for species conservation.

2. A suggestion/question: why use multiple regression to ask whether variation in maximum per-capita growth rate, strength of density-dependence, or both predict mean abundances and population cv? You have a fitted population dynamic model that seems to fit well, at least as far as can be judged by the information provided. Can't you just do a Taylor series expansion around equilibrium population size for each species in an average rainfall year and figure out how the equilibrium population size and the variance around the equilibrium depend on the model parameters alpha, lambda, and beta? That gives you a lot more information than the linear regressions, doesn't it? For instance, it tells about "interaction terms"--how the effects of one model parameter depend on the values of the other parameters. It lets you generalize your conclusions beyond these 6 species to any species to which this model applies. You could solve for rate of return to equilibrium, and prove that it depends on both maximum per-capita growth rate and strength of intraspecific competition. Etc.

We initially chose to use multiple regression because it very tractably demonstrated the combined effects of lambda and alpha on equilibrium abundance, and allowed us to easily relate observed responses and modeled parameters. We are excited about this suggestion, we think it is very interesting. For simplicity we assumed an average rainfall year (i.e., we could ignore beta, as we have standardized). We then realized that, under these conditions and assuming a Poisson distribution, we could analytically solve for both population size and stability.

For equilibrium population size, the results are intuitive, align with our findings, and match our data well. It confirms the joint effect of GRWR/lambda and NDD/alpha on population size. The relationship between lambda and log(equilibrium abundance) is linear, and increases in the value of alpha decrease the slope but do not change the functional form of this relationship.

For population stability, the results largely align with our empirical storyline. Stability increases with GRWR, although this effect is dampened for species with stronger NDD. This makes sense given our assumptions and metric of stability - a higher GRWR increases mean population size and increases recovery when rare, whereas NDD reduces population variability but also reduces population size (the latter of which reduces our stability metric). There was not a strong correlation between predicted and observed stability values. In part, we think this is because climate variability is pronounced at the site, and the potential stabilizing effect of NDD via dampened responses to environmental variability is not reflected in our analytical approach. We think the results of the analytical approach are interesting, and have added discussion and a new figure (Extended Data Fig. 5) accordingly. In addition, we have bolstered our discussion and visualization of the relationship between rainfall variability and stability to better address this dimension of stability.

3. Lines 135-137: I was surprised not to see even brief discussion of the results of Yenni et al. (2017 Global Ecol Biogeog). They use methods similar to those used in this ms to show for 90 communities of various sorts of species (rather than just one plant community) that rare species almost invariably experience stronger negative frequency

dependence than common ones. This implies that persistent rare species are both rare and persistent because they experience much stronger intra- than interspecific competition than common species. That strong intraspecific competition (relative to interspecific competition) both makes rare species rare and helps them bounce back quickly to equilibrium.

Thanks for the suggestion, we have added the following:

“This aligns with recent findings from Yenni et al. (2017), which demonstrate that persistent, rare species are shaped by strong intraspecific NDD that both causes them to be rare but also enables them to rapidly return to equilibrium².”

4. I suggest adding a discussion relating the results to those of Peter Adler's group. They find that in several US perennial plant communities, intraspecific competition is strong relative to interspecific competition. As a result, the responses of species' abundances to abiotic environmental variation don't depend much on interspecific interactions, except when the species within a community differ greatly in their responses to the abiotic environment. See, e.g., Chu and Adler 2015 Ecology and Chu et al. 2016 Nature Comm. And although they don't mention it, it's clear from their data (e.g., Fig. 3 in Chu et al. 2016) that species' abundances depend on both their maximum per-capita growth rates and on the strength of intraspecific competition they experience (relative to the strength of interspecific competition).

Thanks for this. We have added additional references and discussions to Peter's group in a number of places. These include:

This aligns with previous research in our system²⁶ and others²⁷ that suggests strong niche differentiation (i.e., stronger intra- than interspecific competition), and supports a focus on GRWR and se as determinants of abundance.

These dynamics help to explain why we did not observe individual effects of GRWR and NDD on abundance, however other studies suggest they would be discernable with a larger sample size^{6,10}.

Moreover, while we focused exclusively on the direct effects of rainfall, Chu et al. 2016 found that, across species with a range of abundances and demographic parameters, indirect effects of climate variability were also minimized for species with strong intraspecific NDD²⁷.

5. Just a suggestion: instead of Fig. 3, I'd rather see a plot of the temporal variance of each species' abundance vs. the temporal variance of its maximum per-capita growth rate. That seems like a more easily-interpretable way to visualize which species are stable because they're insensitive to rainfall variation, and which species are stable because they are sensitive to rainfall variation but damp out its effects with strong density dependence. Ziebarth et al. (2010 EcoLetts) emphasize the value of this sort of analysis and show that many species seem to have only a weak tendency to bounce back from perturbations.

Thanks for this suggestion, we liked it and gave it a try. The resulting figure is below (on the x axis is the variance in maximum per-capita growth rate, the y is population stability (temporal mean/ temporal std dev). It does make the point that Calycadenia is more stable relative to its how variable its per-capita growth rate is with rainfall than many other species. However, on the whole we would prefer to retain the original figure, because we think it helps to break down the underlying dynamics. In particular, we think the contrasts between panels a and b allow the reader to visually contrast species like Calycadenia (with its steep slope in a and relatively flat slope in b) and Bromus (with its relatively flat slope in a and steeper slope in b). Given that, we like the integration of panels a and b in the regression in c. We have added a one-to-one line to c, and we think this helps the reader visually relate the two more easily. In addition, we have added a panel that directly relates the magnitude of the predicted effect of rainfall on abundance with observed stability.

REVIEWERS' COMMENTS:

Reviewer #1 (Remarks to the Author):

I reviewed the previous version of the manuscript (Rev 1) and was quite excited by the work. Had a few clarifying comments/suggestions. The authors have done well to address those comments. I am pleased with the manuscript as it stands and have no further comments.

Reviewer #2 (Remarks to the Author):

I was reviewer two last time, and I appreciate the fine work the authors have done to incorporate my comments into their revised manuscript. I still find that this is a timely and interesting work. I also had a few more minor comments that I believe would help strengthen the manuscript even more before publication.

Lines 27-29: I get what you are trying to say here, but this statement is not counter intuitive to me. I would certainly expect the factors that govern population dynamics to determine the stability of a species, especially since you control for potential abundance effects by incorporating mean abundance into your measurement of stability (which is fine). Perhaps there could be an alternative wording here that might hit the take-home message a little stronger.

Lines 45-47: I appreciate the change of wording here from "self-limitation" to NDD. However, you may want to clarify that you're referring to conspecific negative density dependence, or negative frequency dependence, which affects conspecifics more than heterospecifics (as you state here) and contributes to the maintains diversity. NDD on its own could also refer to the carrying capacity of all species in a community (regardless of species identity), which does not necessarily affect conspecifics more than heterospecifics and would not necessarily be expected to maintain diversity. I realize that CNDD is not necessarily better here (more jargon), but my concern is that the acronym NDD may confuse readers. I've run into the same problem in my own writing -- it'd be nice if there were a clear, simple and unambiguous term for this very important process.

Lines 49-50: I agree with this line, but a citation should probably be given that cites relatively weak interactions with other species relative to the strength of intraspecific interactions.

** A very minor comment: Lines 40-41: A little awkward having the however in the middle of the sentence here. Obviously your choice of wording here, but I would suggest moving the however to the beginning of the sentence.

Reviewer #3 (Remarks to the Author):

The revised ms addresses my concerns. I'm satisfied that further revisions would not materially improve the ms.

Line-by-line response for Hallett et al. Commonness and rarity of species revisited: relationships with population stability

REVIEWERS' COMMENTS:

Reviewer #1 (Remarks to the Author):

I reviewed the previous version of the manuscript (Rev 1) and was quite excited by the work. Had a few clarifying comments/suggestions. The authors have done well to address those comments. I am pleased with the manuscript as it stands and have no further comments.

Thank you! We really appreciate your previous comments and current support.

Reviewer #2 (Remarks to the Author):

I was reviewer two last time, and I appreciate the fine work the authors have done to incorporate my comments into their revised manuscript. I still find that this is a timely and interesting work. I also had a few more minor comments that I believe would help strengthen the manuscript even more before publication.

Thank you for your helpful comments then and now.

Lines 27-29: I get what you are trying to say here, but this statement is not counter intuitive to me. I would certainly expect the factors that govern population dynamics to determine the stability of a species, especially since you control for potential abundance effects by incorporating mean abundance into your measurement of stability (which is fine). Perhaps there could be an alternative wording here that might hit the take-home message a little stronger.

We appreciate this suggestion and have removed the "counter intuitively".

Lines 45-47: I appreciate the change of wording here from "self-limitation" to NDD. However, you may want to clarify that you're referring to conspecific negative density dependence, or negative frequency dependence, which affects conspecifics more than heterospecifics (as you state here) and contributes to the maintains diversity. NDD on its own could also refer to the carrying capacity of all species in a community (regardless of species identity), which does not necessarily affect conspecifics more than heterospecifics and would not necessarily be expected to maintain diversity. I realize that CNDD is not necessarily better here (more jargon), but my concern is that the acronym NDD may confuse readers. I've run into the same problem in my own writing -- it'd be nice if there were a clear, simple and unambiguous term for this very important process.

Yes, we agree terminology is challenging here. We have compromised by calling it "negative intraspecific density-dependence" when written out, with NDD given as the abbreviation. There have been a number of papers that have used NDD to refer to within-species NDD, and so we hope that our description of the phenomenon plus current usage of NDD in the literature will alleviate confusion.

Lines 49-50: I agree with this line, but a citation should probably be given that cites relatively weak interactions with other species relative to the strength of intraspecific interactions.

Fair enough, we have added a citation to Levine and HillRisLambers 2009.

** A very minor comment: Lines 40-41: A little awkward having the however in the middle of the sentence here. Obviously your choice of wording here, but I would suggest moving the however to the beginning of the sentence.

We think it's fine.

Reviewer #3 (Remarks to the Author):

The revised ms addresses my concerns. I'm satisfied that further revisions would not materially improve the ms.

Thank you!